# Vital Signs Sensing Gown Employing ECG-Based Intelligent Algorithms

**DOI:** 10.3390/bios12110964

**Published:** 2022-11-03

**Authors:** Li-Wei Ko, Yang Chang, Bo-Kai Lin, Dar-Shong Lin

**Affiliations:** 1Center for Intelligent Drug Systems and Smart Bio-Devices (IDS2B), Institute of Bioinformatics and Systems Biology, College of Biological Science and Technology, National Yang Ming Chiao Tung University, Hsinchu 300, Taiwan; 2Institute of Electrical and Control Engineering, Department of Electronics and Electrical Engineering, National Yang Ming Chiao Tung University, Hsinchu 300, Taiwan; 3Drug Development and Value Creation Research Center, Department of Biomedical Science and Environmental Biology, College of Life Science, Kaohsiung Medical University, Kaohsiung 807, Taiwan; 4Department of Biological Science & Technology, National Yang Ming Chiao Tung University, Hsinchu 300, Taiwan; 5Department of Pediatrics, Mackay Memorial Hospital, Taipei 104, Taiwan; 6Department of Medicine, Mackay Medical College, New Taipei 252, Taiwan

**Keywords:** miniaturized circuit, ECG, vital sign, heart rate, respiration, core body temperature, wireless device

## Abstract

This study presents a long-term vital signs sensing gown consisting of two components: a miniaturized monitoring device and an intelligent computation platform. Vital signs are signs that indicate the functional state of the human body. The general physical health of a person can be assessed by monitoring vital signs, which typically include blood pressure, body temperature, heart rate, and respiration rate. The miniaturized monitoring device is composed of a compact circuit which can acquire two kinds of physiological signals including bioelectrical potentials and skin surface temperature. These two signals were pre-processed in the circuit and transmitted to the intelligent computation platform for further analysis using three algorithms, which incorporate R-wave detection, ECG-derived respiration, and core body temperature estimation. After the processing, the derived vital signs would be displayed on a portable device screen, including ECG signals, heart rate (*HR*), respiration rate (*RR*), and core body temperature. An experiment for validating the performance of the intelligent computation platform was conducted in clinical practices. Thirty-one participants were recruited in the study (ten healthy participants and twenty-one clinical patients). The results showed that the relative error of *HR* is lower than 1.41%, *RR* is lower than 5.52%, and the bias of core body temperature is lower than 0.04 °C in both healthy participant and clinical patient trials. In this study, a miniaturized monitoring device and three algorithms which derive vital signs including *HR*, *RR*, and core body temperature were integrated for developing the vital signs sensing gown. The proposed sensing gown outperformed the commonly used equipment in terms of usability and price in clinical practices. Employing algorithms for estimating vital signs is a continuous and non-invasive approach, and it could be a novel and potential device for home-caring and clinical monitoring, especially during the pandemic.

## 1. Introduction

Vital signs are physiological signals that represent the general physical status of an individual. Monitoring vital signs provides a quick and simple evaluation that can be used to understand human body health. The primary four vital signs assessed by medical professionals and health care providers include blood pressure, body temperature, heart rate, and respiration rate [1], which are regularly measured for periodic assessment to hospital inpatients. Once the value of these vital signs falls out of the normal range, the anomaly of the patient can be detected immediately.

Body temperature is a measurement of the ability to generate and get rid of heat from the human body. Since humans are homoeothermic animals, the human body is usually maintained within a stable range of temperature. Body temperature could be categorized into two types: one is skin body temperature, and the other one is core body temperature. Skin body temperature is defined as skin temperature that reflects the state of heat exchange between the human body and a thermal environment [2]. In contrast, core body temperature is defined as the operating temperature of an organism in deep structures of the body, resulting in its immunity to external environmental influences. There are non-invasive and invasive methods to assess core body temperature. The advantage of the invasive method is that it is more accurate, but it is less comfortable. In [3], it is found that using a tympanic thermometer to assess temperature is more accurate than other non-invasive methods, and more comfortable than the invasive method. Therefore, a tympanic thermometer was applied to validate the core body temperature algorithm proposed in this study.

The common method to assess heart rate is to measure the pulse from the radial artery at the wrist [4]. Moreover, heart rate can be calculated from electrocardiography (ECG) with a specific device by identifying the peaks of ECG [5]. The QRS complex is the most obvious waveform in ECG; many studies have developed the QRS detection algorithm to detect QRS complex for assessing heart rate [6,7,8,9]. As for respiration rate, the conventional method to assess it is to count the respiration cycle, i.e., the human chest undulation [10]. However, it is time consuming and inaccurate to calculate the respiration rate with such an approach. Moody et al. [11] provided a method to assess respiration rate from ECG with the “ECG-derived respiration (EDR)”. EDR is defined by the properties of ECG morphology influenced by respiratory-induced chest movements and the changes in the electrical impedance during respiration [12]. Several publications have documented that EDR is reliable for monitoring respiration cycles [13,14,15].

COVID-19 has become a major challenge for public health worldwide since 2019 [16]. Repeated and continuous measurement of body temperature plays an critical role in infection prevention and control of the pandemic [17]. Furthermore, the changes in body temperature also correlate with mortality in COVID-19 patients [18]. Research also found that COVID-19 patients who deteriorate in hospital experience rapidly worsening respiratory failure. In a previous study, Hands et al. [19] proposed warning methods based on monitoring human vital signs. A review paper has also documented that accurate, regular monitoring of vital signs improves recognition of patient deterioration [20]. It is simple to assess a vital sign separately. However, it remains challenging to assess multiple vital signs simultaneously and continuously. Devices in hospitals which are capable of monitoring multiple vital signs continuously are commonly operated by medical specialists, and are neither portable, nor low-priced. The main aim of this study is to design a compact, wireless device which can assess three of the most commonly used vital signs: core body temperature, heart rate, and respiration rate, from only two indicators: skin body temperature and ECG. The device employed QRS detection, EDR, and a novel core body temperature estimation algorithm which were experimentally tested to validate their reliability. With the device and algorithms, this study provides a convenient, comfortable, and affordable method to monitor vital signs constantly and precisely.

## 2. System Architecture

The proposed long-term vital signs monitoring system consists of two components: a vital signs sensing gown and intelligent computation platform (Figure 1). Data acquisition is accomplished in a “vital signs sensing gown”, which is a gown embedded with a novel, compact circuit with two kinds of physiological sensors. The raw data are pre-processed in the circuit and are transmitted to the “intelligent computation platform” for further analysis. After processing, the calculated vital signs would display on a portable device screen including ECG, heart rate (*HR*), respiration rate (*RR*), and core body temperature. Through this long-term monitoring system, users can monitor their ECG signals and vital signs, and especially core body temperature consistently, to know their body condition.

### 2.1. Vital Signs Sensing Gown

The vital signs sensing gown integrates a general gown used in hospitals and a wireless, lightweight, battery-powered miniaturized circuit which acquires two primary physiological signals: ECG (specifically lead II) and skin surface temperature. The circuit is placed in a plastic case to avoid moisture. The wires connected to the electrodes were sewed on the surface of the medical gown. The embedded circuit in the sensing gown was composed of filters, an analog amplifier, an analog-to-digital converter (ADC), and a Bluetooth transmitter module in a compact size, measuring 28 × 28 × 6 mm (Figure 2a). A typical three-lead wireless ECG device was attached to fixed positions of the user’s body via buckles, corresponding to the right arm (RA), left leg (LL), and ground electrodes at the left arm (LA) to collect the signal. The lead II ECG signal was the voltage difference between the LL and RA electrodes, directed from RA toward LL at +60 degrees, yielding the highest R-wave peak among the different limbs’ leads (Figure 2a) [21]. In the miniaturized circuit, the raw ECG signals were sampled at 256 Hz with a 10-bit resolution. The raw data were bandpass filtered from 0.1 to 100 Hz and filtered by a 60 Hz notch filter to remove artifacts including eye blink, muscle movement, and alternating current artifacts. Afterward, the filtered ECG signals were amplified by a magnitude of 800 and converted from analog to digital. The skin surface temperature was acquired by component DS18B20, located on the gown in contact with the skin near the ribs of the human body. The digital thermometer exhibits a ±0.5 °C accuracy ranging from −10 °C to +85 °C. Finally, component HC-06, a Bluetooth module, transmits filtered ECG signals and skin surface temperature values to the Java-based computing platform for further analysis.

### 2.2. Intelligent Computation Platform

The intelligent computation platform is compatible with Bluetooth-enabled portable devices, such as mobile phones and tablets. The filtered ECG signals and skin surface temperature acquired by the vital signs sensing gown were applied into an intelligent computation platform. The outputs of the algorithms were *HR*, *RR*, and core body temperature (Figure 2b). The processing in the platform is instant; therefore, the portable device would display real-time vital sign variation. The refresh rate of *HR* and *RR* are once every six seconds; the value of *HR* is derived by 60 s divided by mean interval of *HR* in these six seconds, and *RR* is derived by multiplying the quantity of *RR* in these six seconds by ten times to obtain the values of *HR* and *RR* per minute. There was storage in the platform so every measurement was recorded and can be accessed anytime. The further description of the intelligent computation platform is introduced in Section 3.

## 3. Integrated Intelligent Algorithm

The integrated intelligent algorithm consists of three main algorithms—R-wave detection, ECG derived respiration, and a novel core body temperature algorithm (Figure 3). The first two algorithms derive *HR* and *RR* from filtered ECG signals, and the core body temperature algorithm derives core body temperature from skin surface temperature and the output of the previous two algorithms.

### 3.1. R-Wave Detection Algorithm

According to the study proposed by Pan and Tompkins [7], *HR* can be derived from R-waves, which are normally the easiest waveforms to identify in ECG signals. In this study, a dynamic threshold method was implemented as the R-wave detection algorithm due to its lower computational requirement [22,23]. The equation of R-wave detection is defined as:(1)$$T\left(n+1\right)=T\left(n\right)+\frac{D\left(n+1\right)-T\left(n\right)}{K}+B,$$
where *n* is a positive integer, *T* is a threshold value, *D* is the present input ECG signal, *K* is the weighting factor, and *B* is the offset value, respectively. According to Equation (1), whenever *D*(*n*+1) is bigger than *T*(*n*+1), the present value of the ECG signal will be considered in the range of R-wave. The algorithm marks the R-peak, which is a local maximum of the R-wave. Figure 4 demonstrates that when the ECG signal (blue line) exceeds the threshold (red line), the maximum value within that period would be regarded as a R-peak. After detecting the R-waves, the algorithm calculates the average time of the *RR* interval (RRI) over six seconds, and the *HR* is derived from 60 divided by the average time of the RRI. For example, if the average RRI is 0.8 s, 60 divided by 0.8 equals 75. Hence, the estimated *HR* would be 75 beats per minute. The input time window is six seconds for the R-wave detection algorithm, so the average RRI would refresh once every six seconds. The displayed output is *HR* with beats per minute as the unit.

### 3.2. ECG-Derived Respiration Algorithm

A study by Arunachalam and Brown [14] demonstrated that R-wave amplitudes are shown to be affected by respiration rhythm. The respiration signal curve can be obtained by the difference in amplitude of the R-wave sequences.
(2)$$a_{n}\left(t\right)\approx \frac{R_{n}\left(t\right)}{\bar{R_{n}\left(t\right)}}$$

Equation (2) is the EDR algorithm, where $a_{n}\left(t\right)$ is the amplitude modulation due to respiratory movement of the thorax, $R_{n}\left(t\right)$ is the composite ECG R-wave amplitude, and $\bar{R_{n}\left(t\right)}$ is the average detected R-wave amplitude. The first step in implementing the EDR algorithm is to find R-peaks derived from the R-wave detection algorithm. The processing flow of the EDR algorithm are as follows:Employ the R-wave detection algorithm to determine R-wave amplitudes.Apply Equation (2) to extract the respiration signals from R-wave amplitude curves.Derive the amount of respiration oscillations to calculate *RR*.

The R-wave detection algorithm was employed to find R-peaks as the data pre-processing for the EDR algorithm. After the R-wave detection algorithm identifies R-peaks, the R-wave amplitudes are recorded and sent to the EDR algorithm. This model applies these amplitudes to extract respiration signals. As the amplitude of the R-peaks changes, the fluctuation can be derived using the values divided by their average (Figure 5). The EDR algorithm extracts R-wave amplitudes every six seconds from ECG signals, and then applies Equation (2) to obtain respiration signals. The displayed output of this algorithm is respiration cycles (breaths) per minute.

### 3.3. Core Body Temperature Algorithm

In this study, an innovative algorithm for estimating core body temperature was proposed. The heat generated inside the body is conducted by blood flow to the skin, where it is dissipated. Skin blood flow correlates to *HR* and is the major mechanism for modulating human body temperature [24]. Another major mechanism for heat regulation is the removal of heat in the water vapor expelled during respiration. Based on these two mechanisms, body temperature, and therefore skin temperature, are affected by *HR* and *RR* [25]. Hypothetically, core body temperature can be estimated by combining chest surface temperature, *HR*, and *RR*. Chest surface temperature can be obtained directly from a digital thermometer, and the other two factors can be determined as previously described. Consequently, the R-wave detection algorithm and EDR algorithm can be used as components to implement core body temperature estimation. The standardized input variables were applied into a multiple linear regression analysis based on the previously mentioned inputs contributing to core body temperature, and used to construct a multiple regression model for core body temperature [26]:(3)$$\mathrm{Core}\;\mathrm{body}\;\mathrm{temperature}={\mathrm{factor}}_{HR\;}\times HR+{\mathrm{factor}}_{RR}\times RR+{\mathrm{factor}}_{T}\times T_{Chest}$$
where factor*_HR_*, factor*_RR_*, and factor*_T_* are generated by multiple linear regression analysis. The values of *HR* and *RR* are acquired by the R-wave detection algorithm and ECG-derived respiration, and *T_Chest_* is the skin surface temperature acquired by DS18B20 measurement, respectively. The processing flow of the core body temperature algorithm works as follows:Calculate *HR* by the R-wave detection algorithm.Calculate *RR* by the EDR algorithm.Apply Equation (3) to implement core body temperature.

The data pre-processing for the core body temperature algorithm is applying the R-wave detection algorithm and the EDR algorithm to obtain *HR* and *RR*, respectively. *HR*, *RR*, and chest surface temperature are three parameters for Equation (3). The core body temperature estimation algorithm on the Java-based computing platform extracts an ECG signal every six seconds, and applies ECG signals to the R-wave detection algorithm and EDR algorithm to obtain *HR* and *RR*. After receiving the values of *HR*, EDR, and chest surface temperature, the core body temperature algorithm derives core body temperature from these three parameters. The displayed output of the algorithm is core body temperature value in Celsius as the unit.

## 4. Experimental Design and Materials

### 4.1. Subjects

Thirty-one volunteers were recruited for this study (seven females and twenty-four males; overall mean age: 26.4 ± 22.0 years; male: 26.0 ± 20.2 years; female: 22.7 ± 26.8 years). Ten healthy participants took part in the healthy participant trials at National Chiao Tung University (one female and nine males; overall mean age 23.6 ± 0.5 years; male: 23.7 ± 0.5 years; female: 23.0 ± 0.0 years). Twenty-one clinical patients took part in clinical patient trials in the hospital. Ten adult patients participated in the experiment at Mackay Memorial Hospital, Taipei Branch (two females and eight males; overall mean age: 54.4 ± 9.7 years; male: 53.1 ± 9.0 years; female: 59.5 ± 14.8 years). Eleven child patients participated in the experiment at Mackay Memorial Hospital, Tamshui Branch (four females and seven males; overall mean age 3.5 ± 1.0 years; male: 3.1 ± 0.9 years; female: 4.3 ± 1.0 years). All the healthy participants did not habitually smoke or drink and had no history of cardiovascular disease. One of the adult patients had the experience of heart surgery, and four of the child patients had a fever during the experiment. All healthy participants and clinical patients were informed of the experimental details and signed the research consent documents prior to experimentation. The procedures of the experiment were approved by the Institutional Review Board (IRB) of the Mackay Memorial Hospital, Taiwan.

### 4.2. Clinical Experimental Design for Algorithm Validation

The experiment was conducted to validate the three algorithms which have been introduced in previous sections: the R-wave detection algorithm, EDR algorithm, and core body temperature algorithm under two different environments. The healthy participants attended the experiment at the National Chiao Tung University, Hsinchu, Taiwan. The clinical patients attended the experiment on the ward at Mackay Hospital, Taipei or Tamshui branch, Taiwan. The participants or patients were instructed to sit in a chair or lie in a bed for twenty minutes. The examiners requested them to be equipped with two devices: the proposed vital signs sensing gown and NeXus-10 to acquire signals from these two devices simultaneously (Figure 6). At the same time, the examiners used a Braun thermometer IRT-4020 to record practical core body temperature repeatedly.

The NeXus-10 (Mind Media) is an equipment approved by the U.S. Food and Drug Administration (FDA). The NeXus-10 obtains an ECG signal by lead II components and acquires a respiration signal by the respiration sensor as a belt set. The respiration sensor is used to monitor abdominal or thoracic breathing to measure breath frequency, which can be worn over clothing. The sampling rate of the ECG signal is at 256 Hz, the as same as our device, and the sampling rate of the respiration signal is at 32 Hz. The Bluetooth module inside the NeXus-10 transmits the acquired signals to computers for further analysis. The Braun thermometer IRT-4020 is also a device approved by U.S. FDA authentication. It was adopted for the validation of the core body temperature algorithm.

## 5. Experimental Results

To evaluate the effectiveness of the algorithms, the data of three devices—vital signs sensing gown, NeXus-10, and Braun thermometer—were analyzed and statistics were gathered to interpret the results.

### 5.1. R-Wave Detection Algorithm Results

To validate the R-wave detection algorithm, relative error was performed to represent the accuracy:(4)$$\mathrm{Relative}\;\mathrm{Error}=\frac{\left|estimated\;value-real\;value\right|}{real\;value}\times 100\%.$$

Relative error is the ratio of absolute error of a measurement to the measurement being taken (Equation (4)). It expresses the relative size of the error of the measurement in relation to the measurement itself. The *HR* values acquired by the R-wave detection algorithm (estimated value) were compared with the values obtained from the NeXus-10 recording (real value).

In both groups of healthy participant and clinical patient trials, the algorithm extracted six seconds of ECG signals to derive *HR*, yielding 200 *HR* values in a total 20 min experiment per subject. The ECG records of two adults and one child in clinical patient trials were discarded due to a data saving problem of the NeXus-10, resulting in 18 data in this dataset. The validation results of two trials of subjects are shown in Table 1 and Table 2. The mean values of healthy participants’ *HR* are shown in Figure 7a. The mean values of clinical patients’ *HR* are shown in Figure 7b. Overall, the relative error in healthy participant trials is 0.45%, and the values of relative error in clinical patient trials are 0.17% and 1.41% for adult group and child group, respectively.

### 5.2. ECG-Derived Respiration Algorithm Results

The *RR* acquired from the EDR algorithm was compared with the NeXus-10 recording which utilized a belt set. In both categories of healthy participant and clinical patient trials, the algorithm extracts six seconds of ECG signals to implement *RR*, and a total 20 min experiment was divided into twenty epochs with *RR* of one minute, resulting in 20 *RR* values per subject. The data of the NeXus-10 from one adult and eight children in clinical patient trials were discarded due to the noise caused by extensive movements of the subjects. The relative error represents the accuracy of the EDR algorithm, the same as for the R-wave detection algorithm. The validation results of two categories of trials are shown in Table 3 and Table 4. The mean values of the 20 epochs in healthy participant trials are shown in Figure 8a. The mean values of the 20 epochs in clinical patient trials are shown in Figure 8b. Overall, the relative error in healthy participant trials is 2.38%, and the relative error in clinical patient trials is 5.52%.

### 5.3. Core Body Temperature Algorithm Results

The values of core body temperature acquired by the novel algorithm were compared with the recorded values of the digital ear thermometer IRT-4020. As the results of multiple linear regression, factor*_HR_*, factor*_RR_*, and factor*_T_* were 0.221, 0.583, and 0.171, respectively. It was noted that the algorithm that estimated body temperature was the summation of the derived value using the factors and the constant of 35.11. The validity of this algorithm was tested by calculating the mean of the root-mean-square-error (RMSE). The RMSE is used to measure goodness-of-fit between values of derived and real core body temperature (Equation (5)). Furthermore, bias is used to evaluate the algorithm if the model underestimates or overestimates the core body temperature (Equation (6)) [27]. The closer the RMSE and bias are to 0, the more precise the proposed method is.
(5)$$\mathrm{RMSE}=\sqrt{\frac{1}{n}\overset{n}{\underset{i=1}{\sum }}{\left(x_{derived}-x_{real}\right)}^{2}}$$
(6)$$\mathrm{Bias}=\frac{\sum^{}\left(x_{derived}-x_{real}\right)}{number\;of\;measurement}$$

Because of a problem of saving data, one child’s data in the clinical patient trial was discarded, resulting in 20 data in this dataset. The validation results of the two categories of trials are shown in Table 5 and Table 6. The mean values in healthy participant trials are shown in Figure 9a. The mean values in clinical patient trials are shown in Figure 9b. Overall, the RMSE and bias in healthy participant trials are 0.04 and −0.04 °C.The RMSE and bias in clinical patient trials are 0.08 and 0.02 °C. Besides statistical results, Figure 10 depicted the comparison of continuous temperature signals between the thermometer recording and the algorithm estimation using one subject as the example in both the healthy participant group and the clinical patient group. The subject in the clinical patient group had a fever, which resulted in the increasing body temperature (Figure 10b). The body temperature of the subject in the healthy participant group remained stable during the time period. However, the derived values with the core body temperature algorithm reflected the ascending trend of the temperature due to the fever.

## 6. Discussion

In this study, a vital signs sensing gown employing three algorithms that derive vital signs including *HR*, *RR*, and core body temperature in real-time was proposed. To derive the values of *HR*, a dynamic threshold approach was implemented for R-wave detection. The low-complexity feature of this approach is compatible with the proposed integrated intelligent algorithm that requires accurate calculation in real-time. A study by Chang et al. [23] demonstrated an automated arrhythmia detection using the dynamic threshold method. The validation results of identifying the R-wave showed the sensitivity of 97.99% on an MIT/BIH arrhythmia database [28]. It concluded that the dynamic threshold provided an accurate detection of QRS in a simple approach. The results of comparing the R-wave detection algorithm estimation and NeXus-10 recording in this study demonstrated the relative error of 0.45 ± 0.64 in healthy participant trials, and 0.17 ± 0.10 and 1.41 ± 2.22 for adults and children, respectively, in clinical patient trials. The relative errors are less than 1% in trials of adults. By contrast, the subjects in younger ages could cause more movement artifacts, resulting in higher relative error in the trials of child patients.

The respiration rate in the system was derived using the EDR algorithm which employed the fluctuations of the R-peak amplitudes. A recent study presented by Varon et al. [12] evaluated 10 different EDR methods, and concluded that the methods based on QRS slopes outperformed the other methods. Furthermore, the simplicity that these methods provided is crucial for development of ambulatory systems. The results of comparing the ECG-derived respiration algorithm estimation and NeXus-10 recording in this study demonstrated the relative error of 2.38 ± 3.58 in healthy participant trials, and 5.52 ± 9.83 in clinical patient trials, which were comparable to the methods based on QRS slopes. However, the proportion of the discarded data in clinical trials of children was high, suggesting that the adopted EDR algorithm might be infeasible for children.

The R-peak detection algorithm and ECG-derived respiration algorithm were proposed by existing research. By contrast, the third algorithm, the core body temperature algorithm, is a novel and self-proposed algorithm. The *HR* and *RR* values derived from the previous two algorithms were recruited and integrated with the skin surface temperature acquired by a digital thermometer to obtain a theoretical core body temperature. According to the validation results, there is no significant difference in *HR*, *RR*, and core body temperature between values measured by current devices (NeXus-10) and values estimated by the algorithms. The core body temperature derived from the core body temperature algorithm is a continuous recording in real-time, allowing the changes of core body temperature to be detected immediately, which is important for ambulatory applications. Once the core body temperature of a patient changes, the medical personnel can respond instantly to avoid certain or potential harm.

There are some important features that describe the vital signs sensing gown.

*Wireless*: The communication between the miniaturized circuit and portable devices is wireless (Bluetooth/Wi-Fi), which enhanced the convenience in applying this device to either healthy participants or clinical patients, inside or outside of hospitals.*Mobility*: The circuit on the sensing gown is small-sized, lightweight, and power-saving compared to current vital signs sensing devices. The proposed device is low-cost, convenient, and most importantly, as accurate as the devices that were commonly used in clinical environments. Furthermore, the circuit is compatible not only with the proposed sensing gown, but with other designs of vital signs sensing systems.*Real-time*: With the simplicity that the proposed algorithms exhibited, the values transmitted by the device and derived from the algorithms can be displayed on portable devices in real-time without any delay.

In conclusion, with the vital signs sensing gown and the intelligent computation platform, three of the important vital signs can be acquired precisely in real-time and in a suitable approach. The core body temperature algorithm provides a modern and accurate method which derives core body temperature through ECG signals and skin surface temperature. Employing these algorithms for estimating core body temperature outperformed the current methods of measuring core body temperature with thermometers, as it is continuous and non-invasive. Exploiting the proposed vital signs sensing gown can be a novel and potential method for home-caring and clinical monitoring, especially during the pandemic.

## 7. Patents

System for detecting core body temperature and method for the same. Patent No.: I650101. 11 February 2019.

## Figures and Tables

**Figure 1 biosensors-12-00964-f001:**
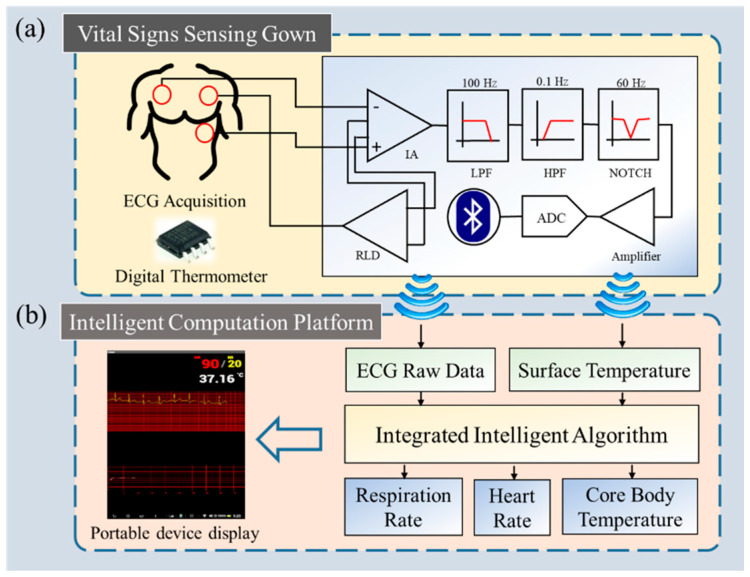
System architecture of long-term vital signs monitoring system. The system consists of two components: (**a**) a sensing gown which acquires lead II ECG signal and surface temperature with the embedded miniaturized circuit, and (**b**) a computation platform. The signals obtained by the gown are pre-processed within, and then transmitted to the computation platform. (RLD, IA, LPF, HPF, NOTCH, and ADC denote right leg driver, instrumentation amplifier, low pass filter, high pass filter, notch filter, and analog-to-digital converter, respectively).

**Figure 2 biosensors-12-00964-f002:**
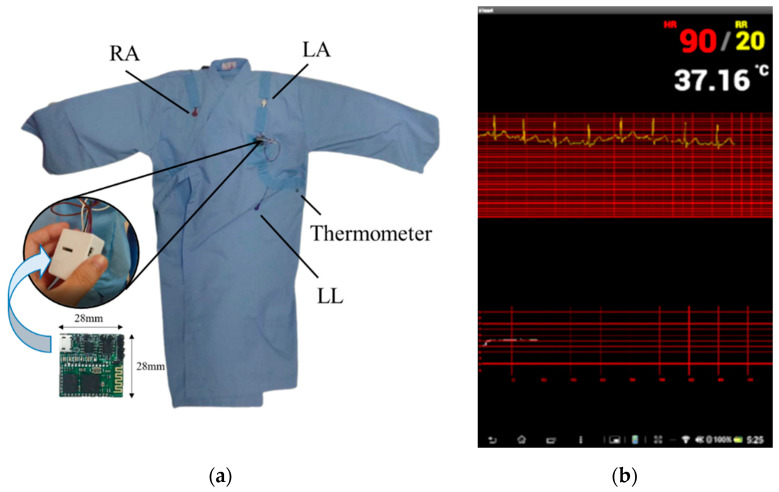
(**a**) Electrode positions on the vital signs sensing gown and placement of the miniaturized circuit. (**b**) Vital signs display on a portable device.

**Figure 3 biosensors-12-00964-f003:**
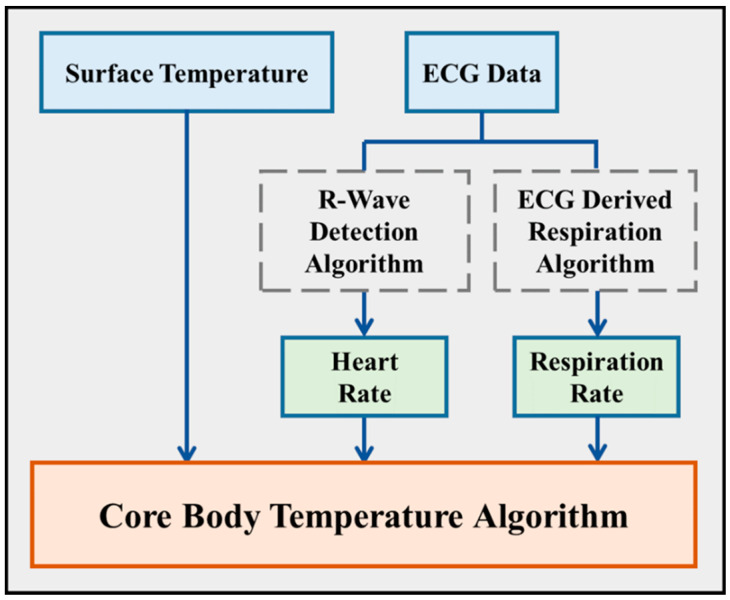
Block diagram of integrated intelligent algorithm. The raw ECG data were applied in two algorithms, R-wave detection algorithm and ECG-derived respiration algorithm, to derive *HR* and *RR*. Later, the values of *HR*, *RR*, and surface temperature were applied in a novel core body temperature algorithm to obtain human core body temperature.

**Figure 4 biosensors-12-00964-f004:**
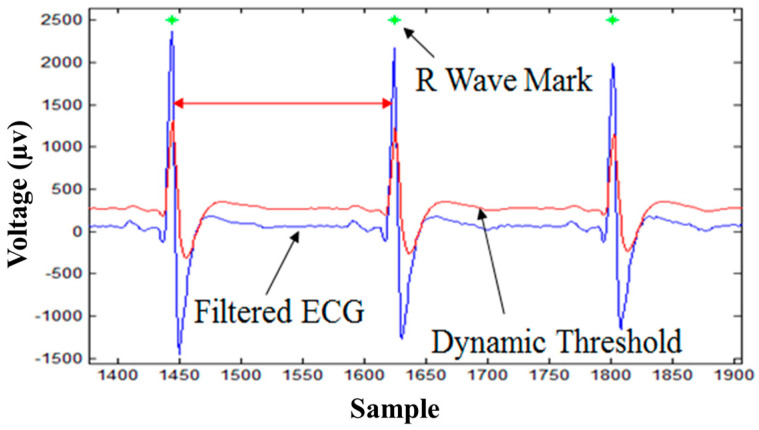
Approach of identifying R-peak using the R-wave detection algorithm. The green star denotes the derived local maximum of the R-wave, which is the R-peak.

**Figure 5 biosensors-12-00964-f005:**
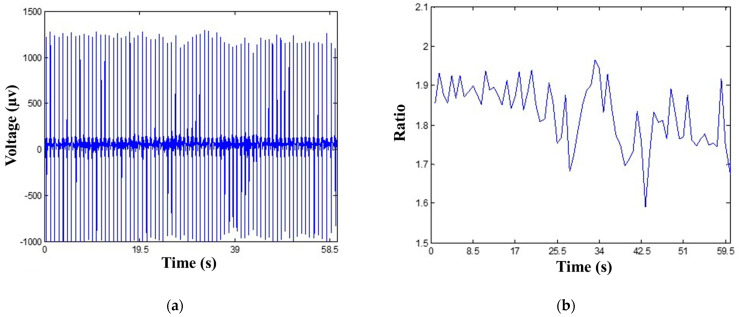
Approach of deriving respiration using the EDR algorithm. (**a**) R-peaks derived by the R-wave detection algorithm. (**b**) Respiratory movement of the thorax derived by the amplitudes of the R-peaks.

**Figure 6 biosensors-12-00964-f006:**
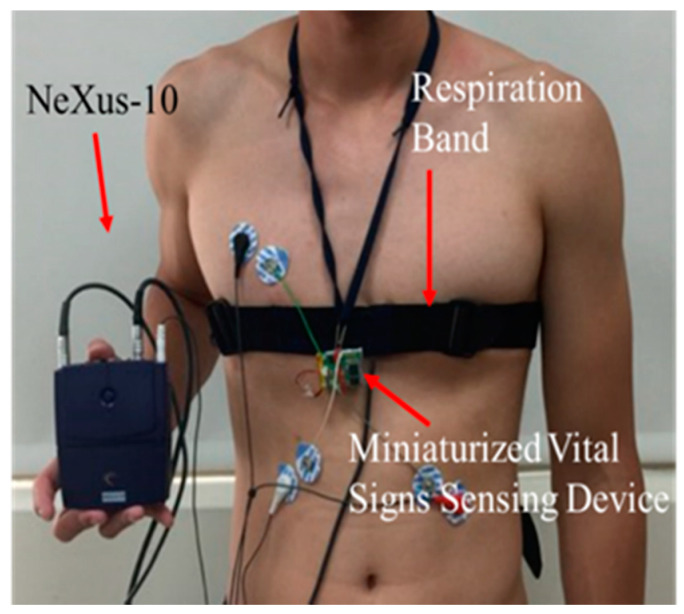
Device setup of experiment for algorithm validation. The respiration sensor is a belt set included in the NeXus-10, which also records ECG signals.

**Figure 7 biosensors-12-00964-f007:**
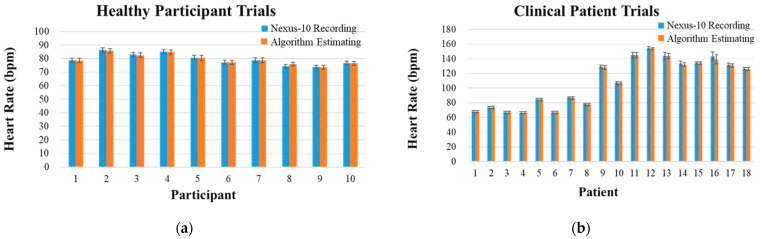
NeXus-10 recording and algorithm estimating values of (**a**) healthy participant trials and (**b**) clinical patient trials for the R-wave detection algorithm. The measurement of *HR* is represented in beats per minute (bpm).

**Figure 8 biosensors-12-00964-f008:**
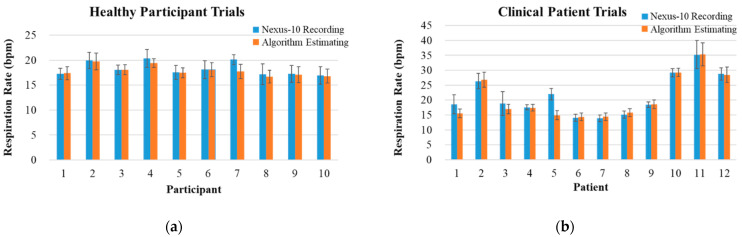
NeXus-10 recording and algorithm estimating values of (**a**) healthy participant and (**b**) clinical patient trials for the ECG-derived respiration algorithm. The measurement of *RR* is represented in breaths per minute (bpm).

**Figure 9 biosensors-12-00964-f009:**
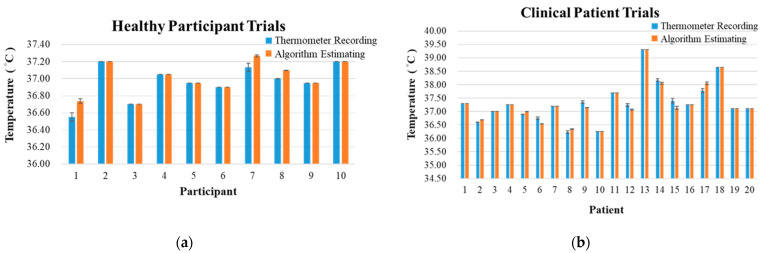
Measured and algorithm estimating values of (**a**) healthy participant and (**b**) clinical patient trials for the core body temperature algorithm. The measurement of core body temperature is represented in degrees Celsius (°C).

**Figure 10 biosensors-12-00964-f010:**
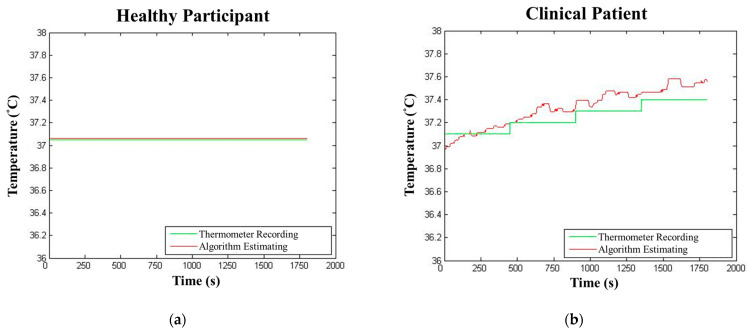
Continuous monitoring of measured and algorithm estimating values of (**a**) a healthy participant and (**b**) a clinical patient for the core body temperature algorithm. The measurement of core body temperature is represented in degrees Celsius (°C).

**Table 1 biosensors-12-00964-t001:** R-Wave detection algorithm results (healthy participant).

	NeXus-10 Recording	Algorithm Estimating
Mean R–R interval ± SD (ms)	757.0 ± 40.3	757.0 ± 37.6
Mean heart rate ± SD (bpm)	79.47 ± 4.28	79.44 ± 3.98
Relative error of heart rate ± SD (%)	0.45 ± 0.64	

**Table 2 biosensors-12-00964-t002:** R-Wave detection algorithm results (clinical patient).

		NeXus-10 Recording	Algorithm Estimating
Mean R–R interval ± SD (ms)	Adults	820.7 ± 87.7	821.5 ± 87.8
	Children	451.6 ± 44.18	458.0 ± 41.7
Mean heart rate ± SD (bpm)	Adults	73.89 ± 8.32	73.81 ± 8.31
	Children	133.87 ± 11.64	131.89 ± 10.78
Relative error of heart rate ± SD (%)	Adults	0.17 ± 0.10	
	Children	1.41 ± 2.22	

**Table 3 biosensors-12-00964-t003:** ECG-derived respiration algorithm results (healthy participant).

	NeXus-10 Recording	Algorithm Estimating
Mean respiration rate ± SD (breaths/min)	18.28 ± 1.34	17.85 ± 1.02
Relative error of respiration rate ± SD (%)	2.38 ± 3.58	

**Table 4 biosensors-12-00964-t004:** ECG-derived respiration algorithm results (clinical patient).

	NeXus-10 Recording	Algorithm Estimating
Mean respiration rate ± SD (breaths/min.)	21.50 ± 7.35	20.56 ± 7.56
Relative error of respiration rate ± SD (%)	5.52 ± 9.83	

**Table 5 biosensors-12-00964-t005:** Core body temperature estimation algorithm results (healthy participant).

	Braun IRT-4020 (Actual Recording)	Algorithm Estimating
Core body temperature ± SD (°C)	36.96 ± 0.21	37.01 ± 0.19
RMSE ± SD (°C)	0.04 ± 0.07	
Bias ± SD (°C)	−0.04 ± 0.07	

**Table 6 biosensors-12-00964-t006:** Core body temperature estimation algorithm results (clinical patient).

	Braun IRT-4020 (Actual Recording)	Algorithm Estimating
Core body temperature ± SD (°C)	37.33 ± 0.74	37.31 ± 0.74
RMSE ± SD (°C)	0.08 ± 0.11	
Bias ± SD (°C)	0.02 ± 0.13	

## Data Availability

Data available on request due to restrictions, e.g., privacy or ethical.

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
