# Peer review of "Vital Signs Sensing Gown Employing ECG-Based Intelligent Algorithms"

_biosensors, 2022, doi:10.3390/bios12110964_

Round 1

Reviewer 1 Report

The authors proposed an integrated wearable sensing system on a gown for monitoring human vital signs including heart rate (HR), respiration rate (RR), and core body temperature, wherein the HE and RR and derived from collected ECG signals using two algorithms and the core body temperature is calculated from the ECG signals and measured skin temperature.  The objective of this research is meaningful, that design a wearable device for human health monitoring.  The experimental steps and results are generally reasonable and well discussed.  The manuscript may be accepted for publication, some concerns/suggestions include:

(1) Only statistical data are presented and analyzed, the real-time signals are also suggested to be provided, particularly the continuous monitoring.

(2) The details/information of the miniaturized circuit is suggested to provide.

(3) Please give the values of the "factors" in the core body temperature algorithm.

(4) The fabrication details of the gown.

(5) What is the big contribution/progress of this project to the field, as numerous wearable electronics are published (and even commercial products) recently.

Reviewer 2 Report

Well, I don't know to which extent crucial data are missing in the paper because of confidentiality of the patent.

Indeed the readers wanting to replicate the experiments for autonomously checking results should need:

1) a better description of the R-Wave detection algorithm (maybe a figure) as equation 1 is only defining the computing of the dynamic threshold but exact definition of how to find the peak of the R-wave (needed for the ECG-Derived Respiration algorithm) is not given

2) a better description of the respiration algorithm should also be provided

3) core body temperature regression analysis used in the paper is different to that used in the cited reference (Niedermann et al., 2014) thus one might be interested in understanding how the factors of the new regression formula were found and which values have been obtained.

I surmise these missing data are because of confidentiality of the patent? Well, understandable, although regrettable!
